# Branched Chain Amino Acids Are Associated with Physical Performance in Patients with End-Stage Liver Disease

**DOI:** 10.3390/biom13050824

**Published:** 2023-05-12

**Authors:** Maria Camila Trillos-Almanza, Hanna Wessel, Magnolia Martínez-Aguilar, Eline H. van den Berg, Rianne M. Douwes, Han Moshage, Margery A. Connelly, Stephan J. L. Bakker, Vincent E. de Meijer, Robin P. F. Dullaart, Hans Blokzijl

**Affiliations:** 1Department of Gastroenterology and Hepatology, University Medical Center Groningen, University of Groningen, P.O. Box 30.001, 9700RB Groningen, The Netherlands; wessel.hanna@gmail.com (H.W.); l.m.martinez.aguilar@umcg.nl (M.M.-A.); e.h.wouters@umcg.nl (E.H.v.d.B.); r.m.douwes@umcg.nl (R.M.D.); a.j.moshage@umcg.nl (H.M.);; 2Labcorp, 3601 Davis Drive, Morrisville, NC 27560, USA; connem5@labcorp.com; 3Department of Internal Medicine, Division of Nephrology, University Medical Center Groningen, University of Groningen, P.O. Box 30.001, 9700RB Groningen, The Netherlands; s.j.l.bakker@umcg.nl; 4Department of Surgery, Division of Hepato-Pancreato-Biliary Surgery and Liver Transplantation, University Medical Center Groningen, University of Groningen, P.O. Box 30.001, 9700RB Groningen, The Netherlands; v.e.de.meijer@umcg.nl; 5Department of Internal Medicine, Division of Endocrinology, University Medical Center Groningen, University of Groningen, P.O. Box 30.001, 9700RB Groningen, The Netherlands; dull.fam@12move.nl

**Keywords:** hepatic cirrhosis, chronic liver disease, liver transplantation, essential amino acids, sarcopenia, hepatic encephalopathy, muscle development, frailty, functional performance, magnetic resonance spectroscopy

## Abstract

Decreased circulating branched chain amino acids (BCAA) represent a prominent change in amino acid profiles in patients with end-stage liver disease (ESLD). These alterations are considered to contribute to sarcopenia and hepatic encephalopathy and may relate to poor prognosis. Here, we cross-sectionally analyzed the association between plasma BCAA levels and the severity of ESLD and muscle function in participants of the liver transplant subgroup of TransplantLines, enrolled between January 2017 and January 2020. Plasma BCAA levels were measured by nuclear magnetic resonance spectroscopy. Physical performance was analyzed with a hand grip strength test, 4 m walking test, sit-to-stand test, timed up and go test, standing balance test and clinical frailty scale. We included 92 patients (65% men). The Child Pugh Turcotte classification was significantly higher in the lowest sex-stratified BCAA tertile compared to the highest tertile (*p* = 0.015). The times for the sit-to-stand (r = −0.352, *p* < 0.05) and timed up and go tests (r = −0.472, *p* < 0.01) were inversely correlated with total BCAA levels. In conclusion, lower circulating BCAA are associated with the severity of liver disease and impaired muscle function. This suggests that BCAA may represent a useful prognostic marker in the staging of liver disease severity.

## 1. Introduction

End-stage liver disease (ESLD) is the final stage of a chronic liver injury, affecting people in their most productive years of life. It represents a leading cause of death worldwide [1], whose only curative treatment, orthotopic liver transplantation (OLT), requires lifelong immunosuppressive therapy and remains cumbersome due to the limited access to donor organs. Patients with ESLD may have impaired physical and neurological functions that severely affect their functional status, resulting in an increased risk of decompensation, wait-list mortality and poorer outcomes after OLT [2,3,4,5].

The two most commonly used scoring systems to determine the severity and prognosis of patients with chronic liver disease are the Child Pugh Turcotte (CPT) and the Model for End-stage Liver Disease (MELD) scores [6,7]. Despite the fact that these scoring systems are commonly accepted as objective predictors of survival in patients with chronic liver disease, both have shortcomings, obviously impacting their ability to predict prognosis. The main limitation of the CPT score is the variation caused by the subjective estimation of the severity of ascites and hepatic encephalopathy (HE). Even though the MELD score is not subject to variation in individual interpretation, it may not predict prognosis and mortality with sufficient accuracy in patient groups suffering from frequently coinciding complications [8].

The branched chain amino acids (BCAA), leucine, isoleucine and valine, constitute approximately 35% of essential amino acids [9]. They have protein anabolic properties, promote glucose transport and affect gene expression and hepatocyte cell cycle pathways [10,11]. Imbalances in amino acid metabolism have been described in patients with chronic liver disease [12], which are primarily characterized by a decrease in BCAA levels and an increase in aromatic amino acids (AAA), resulting in a decreased ratio of BCAA to AAA, called the Fischer ratio [10]. With the progression of cirrhosis, the Fischer ratio tends to be lower and may be regarded as an indicator for nutritional status and the prognosis of cirrhotic patients [10]. This decrease in BCAA in patients with ESLD is reverted after OLT [13].

Overall, while the exact mechanisms underlying the pathogenic importance of low BCAAs in liver disease are not fully understood, there is evidence to suggest that decreased BCAA levels may play a role in the development and progression of liver disease through various mechanisms, including impaired protein synthesis and turnover, utilization in skeletal muscle, increased inflammation, and the dysregulation of signaling pathways [14,15,16]. One of the proposed causes of this amino acid deficiency is the enhanced consumption of BCAA for ammonia detoxification to glutamine in muscles, likely resulting in HE and sarcopenia [17].

We hypothesized that low circulating BCAA concentrations could be associated with poor physical performance in ESLD. The present study was initiated to investigate the relationship between circulating concentrations and physical performance, as well as CPT and MELD scores in patients with ESLD.

## 2. Materials and Methods

### 2.1. Study Population

A cross-sectional, single-center analysis from the University Medical Center Groningen (UMCG) was conducted using the data from patients enrolled in the prospective observational cohort study and solid-organ transplantation biobank “TransplantLines” (NCT02811835) [18]. TransplantLines was approved by the Medical Ethics Committee of the University Medical Center Groningen (METc 2014/077) and is performed in accordance with the guidelines of the Declaration of Helsinki [19]. All participants provided written informed consent. Patients with cirrhosis (aged > 18 years) eligible for transplantation who completed the screening program for liver transplantation from January 2017 until January 2020 were included. Patients without cirrhosis, with no mastery of the Dutch language and with an inability to intellectually understand questionnaires or physical tests and those with missing BCAA values were excluded. None of the patients received supplementation with BCAA.

### 2.2. Laboratory Analysis

Laboratory measurements were extracted from the biobank database including total cholesterol, triglycerides, glucose, HbA1c, hemoglobin, albumin, C-reactive protein (CRP), aspartate transaminase (AST), alanine transaminase (ALT), alkaline phosphatase (ALP), total bilirubin, platelet counts, leucocyte counts, serum creatinine, ammonia and urinary creatinine excretion and clearance. In addition, ethylene diamine tetraacetic acid (EDTA)-anticoagulated plasma samples were taken from the biobank and stored at −80 °C until further analysis.

Plasma concentrations of BCAA and alanine were measured using a Vantera Clinical Analyzer at Labcorp (Morrisville, NC, USA), a fully automated, high-throughput, 400 MHz proton (^1^H) nuclear magnetic resonance (NMR) spectroscopy platform with Agilent Technologies [20]. In brief, plasma samples were prepared on board the instrument by mixing 1:1 sample: NMR diluent (50 mM sodium phosphate, 120 mM KCl, 5 mM Na_2_EDTA, 1 mM CaCl_2_, pH 7.4) and automatically delivered to the flow probe in the spectrometer’s homogeneous magnetic field. Data acquisition on the Vantera was accomplished with water suppression using the WET solvent suppression technique [20,21].

In the ^1^H NMR spectrum, the methyl signals from the three BCAA exhibit distinct patterns that can be used for quantification. An optimized deconvolution algorithm was developed to mathematically model the methyl signals from BCAAs (between 0.718 and 1.02 ppm) in each NMR spectrum, allowing for the quantification of valine, leucine, isoleucine and alanine. The concentrations of BCAAs were determined using non-negative linear least squares by first determining the signal areas of their well-characterized peaks pattern and then multiplying by predetermined conversion factors (micromole units) [20,22].

Total BCAA concentrations were calculated as the sum of valine, leucine, and isoleucine. The coefficients of variation for inter- and intra-assay precision were as follows: total BCAA (1.8–6.0%), valine (1.7–5.4%), leucine (4.4–9.1%), isoleucine (8.8–21.3%) and alanine (1.3–2.8%).

### 2.3. Severity of Liver Disease

To assess the severity of liver disease, CPT and MELD scores were used. The CPT score was calculated by adding up points according to the following scheme: total bilirubin < 35 µmol/L = 1, 35–51 µmol/L = 2, >51 µmol/L = 3; Albumin > 35 g/L = 1, 28–35 g/L = 2, <28 g/L = 3; INR < 1.7 = 1, 1.7–2.2 = 2, >2.2 = 3, ascites absent = 1, slight = 2, moderate = 3; HE absent = 1, slight = 2, moderate = 3. The MELD score was calculated according to the formula: (0.957 ∗ ln(Serum Cr) + 0.378 ∗ ln(Serum Bilirubin) + 1.120 ∗ ln(INR) + 0.643) ∗ 10 [23]. The presence of ascites was established with imaging (ultrasound, computed tomography or magnetic resonance imaging). The grade of varices was determined by endoscopy (grade 1 small varices, minimally elevated veins above the mucosal surface; grade 2 medium varices, tortuous veins occupying < one-third of lumen; grade 3 large varices occupying > one-third of lumen). HE was defined according to the West Haven criteria [24]. Hepatorenal syndrome was defined according to the International Ascites Club’s Diagnostic Criteria [25].

### 2.4. Metabolic Measurements

A diagnosis of type 2 diabetes mellitus (T2DM) was confirmed when a subject had self-reported T2DM, used glucose-lowering medication (oral agents and insulin), had a fasting glucose (FG) > 7.0 mmol/L and/or had a non-fasting glucose > 11.0 mmol/L or had an HbA1c > 47.5 mmol/mol (6.5%).

Blood pressure (mmHg) was measured according to a standard clinical protocol using an automatic device (Philips SureSign VS2+, Andover, MA, USA). It was measured four times in the sitting position, preferably on the right arm and with 3 min between each measurement. During measurements, patients were on their regular medication, including antihypertensive drugs (calcium blockers, beta blockers, potassium sparing diuretics, ace inhibitors, ARBs, loop diuretics, thiazide diuretics), and high blood pressure was defined as blood pressure ≥ 130/85 mmHg. The median of the four blood pressure measurements was used for further data analysis.

### 2.5. Sarcopenia and Frailty

To evaluate physical performance, five tests were used: a hand grip strength, standing balance test, 4 m walking test, sit-to-stand test and timed up and go performance test (TUG). Hand grip strength was measured on the dominant hand using the Jamar Hydraulic Hand Dynamometer (Patterson Medical JAMAR 5030J1, Warrenville, QC, Canada) in the sitting position, with arms in 90-degree flexion. Participants were asked to perform a maximal isometric contraction. The measurement was repeated three times, with 30 seconds of recovery between each measurement, and the mean hand grip strength was used for further analysis [17]. The standing balance test was performed on the ground with feet next to each other, opened eyes and crossed arms. The second attempt was with closed eyes. During the third and fourth attempts, the patient was standing on a mat, and the last attempt was on the floor with crossed feet and open eyes. The 4 m walking test consisted of walking along a straight line of 4 m. The test was performed twice, and the averaged time was calculated. In the sit-to stand test, patients were instructed to stand up five times from a sitting position as fast as possible. The initial position was sitting on a chair in an upright position with arms folded across the chest. This test was repeated three times after a prior test round, and the average time in seconds was calculated. For the TUG test, a pylon and a chair were put 3 m apart. The test was performed three times, with a trial round before where no time was measured. Participants were asked to stand up from the chair without using their arms, walk with normal speed around the pylon, go back to the chair and sit down again. Participants with a walking aid were permitted to use it during the test. Time was measured in seconds, and the average time was used for further analysis.

To assess frailty, the Clinical Frailty Scale (CFS) was scored at study visits by a trained investigator. The CFS is a validated frailty measurement, and frailty is scored based on clinical judgment on a continuous scale from 1 (very fit) to 9 (terminally ill). A CFS score of ≥5 is generally considered frail [18].

### 2.6. Statistical Analysis

Data analysis was performed with IBM SPSS software (SPSS, version 25.0, SPSS Inc., Chicago, IL, USA). Continuous variables were expressed as the mean ± SD for normally distributed variables and as the median (interquartile range) for non-normally distributed variables. Between-group variables were analyzed using one-way ANOVA for unpaired normally distributed data. Non-normally distributed data were log-transformed to achieve normal distribution. Fisher’s exact test was used for the analysis of categorical data, when appropriate. Univariate correlations for continuous response variables were analyzed by Pearson correlation coefficients. Two-sided *p*-values < 0.05 were considered to be statistically significant.

## 3. Results

Baseline characteristics divided according to sex-stratified BCAA tertiles are shown in Table 1. A total of 92 subjects (65% men and 35% women) with cirrhotic ESLD were included. The mean age of the patient population was 56.8 ± 9.7 years. The median concentration of total BCAA was 307.0 (289.5–356.8) µmol/L in men and 213.5 (211.1–274.5) µmol/L in women (*p* < 0.001). These BCAA concentrations are much lower than those observed in the population-based Prevention of Renal and Vascular End-stage Disease (PREVEND) cohort study, corresponding to 412.7 (231.4–1042.9) µmol/L in men and 339.7 (184.5–711.4) µmol/L in women (*p* < 0.001 for each gender) [26].

Patients in the lowest BCAA tertile had higher CPT scores and were more likely to have alcoholic liver disease (Table 1). Other variables such as age, smoking status, body composition, blood pressure, heart rate and the use of medication were not different between the groups. High blood pressure and T2DM were present in 63 and 50 patients (*p* = 1. *p* = 0.26), respectively. Liver disease complications (variceal bleeding, hepatocellular carcinoma and hepatorenal syndrome) were also not different between the BCAA tertiles (Table 1). Plasma BCAA concentrations were not significantly different in patients with or without high blood pressure (275 (138–982) µmol/L vs. 278 (145–526) µmol/L, *p* = 0.80), T2DM (325 (159–594) µmol/L vs. 261 (138–982) µmol/L, *p* = 0.09) or MAFLD (287 (138–594) µmol/L vs. 274 (147–982) µmol/L, *p* = 0.96). Only viral hepatitis and alcoholic cirrhosis were related to BCAA plasma levels. Triglycerides were higher and ammonia was lower in the highest BCAA tertile. Complications such as HE, ascites and the use of lactulose were associated with lower BCAA tertiles.

Disease severity, as determined by the CPT score but not significantly so by the MELD score, was worse with lower total BCAA (Table 1). Hence, the total and individual BCAA were analyzed in the different CPT categories. Table 2 shows that BCAA were the highest at the lowest CPT category (A), with valine, leucine and isoleucine each being significantly different. In patients with decompensated liver disease (CPT B and C), total and individual BCAA were lower compared to those in patients in the CPT A group. Wait-list mortality was significantly higher in the second compared to the third total BCAA tertile (Table 1). Alanine did not significantly vary according to the CPT category.

In order to ascertain whether the statistical outcomes pertained to the liver disease stage or potential confounding variables, such as drugs and comorbidities, we performed multivariable linear regression analysis (Appendix A). None of the drug classes used were associated with plasma BCAA concentrations, and only the presence of T2DM was associated with plasma BCAA concentrations (β = 0.222, *p* = 0.036).

Next, we analyzed the concentration of BCAA and their relationship with physical activity by using different exercises such as the hand grip strength test (n = 77), 4 m walking test (n = 42), sit-to-stand test (n = 40), TUG test (n = 40), standing balance test (n = 36) and CFS (n = 80) (Table 3). Note that a considerable number of patients were unable to perform these tests due to a poor clinical condition, which is illustrated by a more than twofold higher mortality on the wait-list compared to patients who performed the tests (33.3% vs. 12.5%; data not shown). Hand grip strength varied significantly between sexes, it being significantly lower in women (38.0 ± 7.9 kg in men vs. 22.1 ± 5.2 kg in women; *p* < 0.001). Hand grip strength was lower in men with the lowest BCAA but not in women (Table 3). The 4 m walking test, standing balance test and CFS were not significantly different between BCAA tertiles (Table 3). In contrast, the times for the sit-to-stand test and TUG test were significantly longer in patients in the lower BCAA tertiles compared to patients in the highest tertile (Table 3).

Univariate regression analysis showed that hand grip strength was related to leucine and isoleucine in men but not in women (Table 4). There were inverse correlations of the TUG performance test with total BCAA, valine and isoleucine. In addition, total BCAA, valine and leucine were inversely related in the sit-to-stand test. Total BCAA and the individual BCAA were unrelated to the 4 m walking test, standing balance test and CFS (Table 4).

In linear regression analysis, the muscular function tests were not significantly associated with the MELD score (hand grip strength, r = 0.136, *p* = 0.238; 4 m walking test, r = 0.223, *p* = 0.155; sit-to-stand test, r = 0.307, *p* = 0.054; TUG test, r = 0.186, *p* = 0.154) and the sit-to-stand test (r = 0.268, *p* = 0.094). CFS was significantly correlated with the muscular function tests (hand grip strength, r = 362, *p* = 0.002; 4 m walking test, r = 0.502, *p* = 0.001; TUG test, r = 0.341, *p* = 0.019). Furthermore, CFS was associated with MELD and CPT (r = 0.325, *p* = 0.003 and r = 0.312, *p* = 0.005, respectively). Additionally, we performed multivariable linear regression analysis including the CPT category, MELD score and plasma BCAA to predict physical performances (Appendix A). Our findings showed that with respect to the hand grip strength test, only the contribution of BCAA was statistically significant when compared to the CPT and MELD scores. The association with BCAA was also significant for the TUG test.

Finally, we performed a sensitivity analysis with alanine to disclose whether the findings were limited to BCAA. Notably, liver function tests, different causes of liver disease as well as complications of liver disease were not significantly different between sex-stratified alanine tertiles (Table 5). The CPT categories and MELD score were also not significantly different between alanine categories (Table 5).

## 4. Discussion

In this study, we found that decreased plasma levels of BCAA were associated with the severity of liver disease, as well as with muscle function. These findings emphasize the relevance of BCAA as a prognostic and severity marker in staging ESLD. We compared plasma BCAA levels in the current ESLD patients with those derived from the general population and found much lower BCAA levels in both men and women with ESLD.

The MELD score is a widely accepted tool for assessing prognosis in patients with ESLD; nevertheless, it has its limitations, mainly because it does not include complications of liver disease including HE [8]. Furthermore, while mortality on the wait-list for OLT has decreased since the use of MELD scores in clinical practice, an increased mortality post-OLT has been noted [27]. Those findings drive the debate on whether the MELD score is accurate enough to predict the prognosis of patients on the wait-list [28]. As BCAA were associated with both HE and sarcopenia, we hypothesized that they might have additional prognostic value in assessing the severity of liver disease. Although the CPT score also takes HE into account, the advantage of BCAA as a biomarker is that it lacks subjective interpretation and may also reflect the presence of sarcopenia due to the association found.

Of potential interest, our study showed that wait-list mortality was lower in patients in the third BCAA tertile, along with worse CPT categorization, although there was no significant difference between MELD scores across BCAA tertiles. This could be related to the fact that some patients with decompensated ESLD had low MELD scores.

Due to their involvement in protein metabolism, low BCAA levels have been proposed to play a causal role in the development of sarcopenia. In the current study, the results of muscular function tests (TUG and sit-to stand test) were associated with BCAA levels. These tests are more complex in comparison to the other muscular function tests (hand grip strength and 4 m walking test), as they combine strength, coordination and balance. So, they could more accurately reflect muscle function. Muscular function tests were not associated with the MELD score and did not predict mortality. These apparently unexpected findings are likely explained by the fact that only a subgroup of patients were physically capable of performing the tests, which led to bias in patient selection, resulting in a relatively healthier subgroup of patients that were capable of performing the tests. In fact, mortality in the patient group that did not perform the timed up and go test was twofold higher.

BCAA take part in the process of ammonia detoxification, and this metabolic step occurs in skeletal muscle, which explains at least in part the close association between HE and sarcopenia [29]. In the current study, we demonstrated a close association between the occurrence of HE and plasma BCAA values. Patients within the lowest BCAA tertile had a greater than twofold increase in the risk of having HE, and patients in the highest BCAA tertile had the lowest ammonia levels, which may suggest that muscular BCAA uptake increases with rising ammonia concentrations in decompensated patients. Although we were unable to determine the role of BCAA in detoxifying ammonia directly, it is conceivable that BCAA alterations may reflect the effectiveness of the detoxification process. This idea is in line with earlier observations, which may suggest that muscular BCAA uptake increases with rising ammonia concentrations in decompensated patients [30]. Such a mechanism needs to be investigated further.

Sarcopenia and HE are both predictors of mortality in patients with ESLD, independent of MELD scores, showing that these two complications have additional prognostic value for this score [31]. A previous study showed that the wait-list survival was shorter in liver disease patients suffering from sarcopenia; this effect was even more pronounced in patients with lower MELD scores [32]. Furthermore, the addition of sarcopenia to the MELD score improved its predictive ability [33]. Therefore, we postulate that including BCAA measurements into the prognostic classification could improve outcome determination before and after OLT and even identify patients who would benefit from supplementation, thereby improving overall survival.

Until now, focus has been largely on the effect of the supplementation of BCAA on various outcome measures. Interestingly, we found that the patient group with the highest tertile had BCAA concentrations comparable to patients without liver disease recruited from the general population, with an average value of about 370 µmol/L in men and women combined [26,34]; they also had the lowest wait-list mortality. These findings suggest that not every patient with ESLD needs BCAA supplementation. Future research should focus on the effect of low BCAA levels on postoperative complications and the appropriate dose of its supplementation to improve outcomes in certain groups of patients.

In multivariable linear regression analysis, we found a significant association of BCAA with T2DM. Diabetic patients were included because excluding participants with comorbidities would, to a major extent, limit the generalizability of the findings of our study. Reassuringly, the prevalence of comorbidities (high blood pressure and T2DM) and MAFLD was not different across the BCAA tertile groups. In comparison, in a cohort of post-liver-transplantation patients, we documented recently that the presence of T2DM was more frequent among patients in the highest tertile of plasma BCAA concentrations [13]. Likewise, that study also showed a 1.60-fold increased odds ratio for T2DM per one standard deviation increase in plasma BCAA [13], underscoring the relevance of the diabetic state in BCAA metabolism [26]. Of further note, the prevalence of alcoholic cirrhosis was highest in the lowest tertile of plasma BCAA. In comparison, early studies revealed lower plasma BCAA in alcoholic fatty liver disease [35] and a lower BCAA/Tyrosine ratio in patients with compensated and decompensated alcoholic cirrhosis [36]. Such an effect of alcohol was also observed after single alcohol administration in healthy subjects [36]. Only a few patients in our study had cirrhosis due to viral infection, with the highest prevalence apparently being observed in patients with the highest tertile of plasma BCAA. The relevance of this finding is uncertain, also taking the progressively lower BCAA/Tyrosine ratio with more severe liver cirrhosis due to hepatitis C into consideration [37]. Finally, it should be mentioned that about 29% of the whole ESLD cohort had concomitant hepatocellular carcinoma, with the highest prevalence in the tertile of patients with the highest plasma BCAA concentration. As a responsible mechanism, the Warburg effect, which is a high degree of glucose uptake and glycolysis followed by lactic acid fermentation, could be operative in carcinoma tissue, possibly contributing to the relative maintenance of circulating BCAA in cirrhosis complicated by hepatocellular carcinoma [38]. On the other hand, BCAA supplementation may have a beneficial effect on hepatocellular carcinoma progression [16].

The current study has several strengths. This is the first cross-sectional analysis of associations between BCAA and muscle function in patients with ESLD wait-listed for transplantation. Furthermore, the patients from the TransplantLines cohort study are well characterized, with extensive data on clinical endpoints. Moreover, alanine did not vary considerably with CPT and MELD scores, which indicates that BCAA and alanine are not, to a major extent, uniformly affected by, or in association with, severe liver disease. Several limitations of this study also need to be considered. First, we did not measure aromatic amino acids in our cohort, which are increased in ESLD patients [12,39,40]. Second, only some of the included patients were able to perform muscular function tests, leading to selection bias and likely the underestimation of the physical performance outcomes.

## 5. Conclusions

In conclusion, the use of high-throughput NMR methodology to determine decreased plasma levels of BCAA may have potential as an indicator for the progression and prognosis of ESLD due to its association with physical performance. Further research is necessary to delineate its clinical relevance, given the limited sample size and cross-sectional design of our study.

## Figures and Tables

**Table 1 biomolecules-13-00824-t001:** Baseline characteristics of 92 patients with cirrhotic ESLD, divided into sex-stratified plasma total BCAA concentrations.

	Total	T1	T2	T3	*p*-Value
Men (n)	60	20	20	20	
Total BCAA (µmol/L)	307.0 (289.5–356.8)	216.0 (192.9–233.0)	307.0 (295.0–314.3)	423.0 (385.8–517.8)	<0.001
Valine (µmol/L)	157.5 (149.8–187.6)	110.5 (99.7–116.2)	157.5 (151.2–161.4)	222.5 (204.1–279.5)	<0.001
Leucine (µmol/L)	98.0 (88.0–115.9)	55.5 (49.7–64.2)	98.0 (92.1–100.7)	130.5 (123.4–181.8)	<0.001
Isoleucine (µmol/L)	52.0 (47.8–57.2)	34.0 (30.3–36.7)	52.0 (50.3–54.1)	67.5 (65.5–78.3)	<0.001
Alanine (µmol/L)	297 (284.0–342.1)	202.5 (181.4–212.1)	297.0 (280.3–313.5)	426.5 (416.0–475.1)	<0.001
Women (n)	32	11	10	11	
Total BCAA (µmol/L)	213.5 (211.1–274.5)	174.0 (160.6–181.6)	213.5 (202.4–223.4)	334.0 (288.8–394.471)	<0.001
Valine (µmol/L)	117 (112.9–146.4)	84.0 (81.0–96.1)	117.0 (111.8–127.2)	177.0 (152.0–208.0)	<0.001
Leucine (µmol/L)	61.0 (57.2–82.6)	36.0 (32.0–43.4)	61.0 (55.6–66.6)	114.0 (91.6–128.6)	<0.001
Isoleucine (µmol/L)	37.5 (36.2–50.1)	28.0 (23.5–30.7)	37.5 (35.0–39.6)	61.0 (53.1–76.0)	<0.001
Alanine (µmol/L)	243.5 (234.2–334)	176.0 (138.8–200.6)	243.5 (230.8–256.4)	426.0 (350.7–519.6)	<0.001
Demographics					
Age at screening	56.8 ± 9.7	58.2 ± 7.8	56.6 ± 9.9	55.7 ± 11.3	0.59
Current smoker, n (%)	20 (21.7)	9 (29.0)	4 (13.3)	7 (22.6)	0.37
Anthropometry					
Height (cm)	174.5 ± 9.6	174.9 ± 7.7	175.0 ± 9.34	173.6 ± 11.7	0.82
Weight (kg)	85.4 ± 18.6	88.0 ± 17.7	84.7 ± 17.0	83.6 ± 21.2	0.63
BMI (kg/m^2^)	27.9 ± 4.9	28.7 ± 5.2	27.6 ± 4.6	27.4 ± 5.1	0.52
Comorbidities, n (%)					
High blood pressure	63 (68.5)	21 (67.7)	21 (70.0)	21 (67.7)	1.000
Diabetes	50 (58.8)	13 (46.4)	19 (67.9)	18 (62.1)	0.26
Circulation					
Heart rate (bpm)	73 ± 12	74 ± 10	71 ± 14	74 ± 12	0.45
SBP (mmHg)	120 ± 20	122 ± 20	116 ± 17	124 ± 21	0.25
DBP (mmHg)	66 ± 11	65 ± 12	63 ± 10	69 ± 11	0.06
Laboratory measurements					
Total cholesterol (mmol/L)	3.6 ± 1.4	3.5 ± 1.2	3.1 ± 1.2	3.9 ± 1.7	0.06
Triglycerides (mmol/L)	1.1 (0.7–1.4)	0.8 (0.6–1.3)	1.0 (0.7–1.4)	1.2 (1.0–1.7)	<0.001
Glucose (mmol/L)	8.2 ± 6.3	7.1 ± 2.5	7.7 ± 2.9	9.6 ± 10.0	0.52
HbA1c (mmol/mol)	33.5 ± 11.8	30.8 ± 7.6	33.2 ± 9.6	36.9 ± 16.4	0.19
Hemoglobin (mmol/L)	6.9 ± 1.3	6.7 ± 1.4	6.8 ± 1.1	7.3 ± 1.3	0.12
Albumin (g/L)	32.7 ± 6.5	31.2 ± 5.3	31.7 ± 5.6	35.1 ± 7.8	0.04
CRP (mg/L)	9.5 (4.2–26.5)	8.0 (3.2–23.0)	11.0 (5.7–28.8)	7.0 (3.9–32.0)	0.63
AST (U/L)	54.0 (43.3–83.0)	48.0 (43.0–65.0)	54.5 (45.0–85.8)	58.0 (37.0–83.0)	0.44
ALT (U/L)	39.5 (28.3–55.8)	31.0 (25.0–48.0)	41.0 (31.5–55.3)	43.0 (33.0–63.0)	0.04
ALP (U/L)	153.0 (156.7–197.3)	150.0 (147.7–233.5)	168.0 (149.9–201.0)	142.0 (132.6–202.3)	0.62
Bilirubin (μmol/L)	41.0 (18.3–86.0)	39.0 (17.0–57.0)	48.0 (22.8–91.0)	30.0 (10.0–111.0)	0.56
Thrombocytes (10^9^/L)	121.4 ± 63.8	127.0 ± 66.0	112.6 ± 67.1	124.4 ± 59.0	0.65
Leucocytes (10^9^/L)	5.6 ± 3.3	6.7 ± 4.1	5.0 ± 3.3	5.2 ± 1.9	0.09
Serum creatinine (µmol/L)	77.0 (62.0–100.0)	70.0 (61.0–118.0)	88.0 (69.0–101.0)	72.0 (59.0–84.0)	0.21
Creatinine excretion (mmol/24 h)	8.2 ± 3.8	8.1 ± 3.3	8.6 ± 4.3	7.9 ± 3.8	0.76
Creatinine clearance (mL/min)	89.8 (56.5–122.2)	77.0 (55.0–125.0)	89.1 (49.3–115.5)	102.3 (81.0–123.2)	0.21
Ammonia (µmol/L)	68.8 ± 33.4	78.1 ± 34.5	75.5 ± 37.2	51.6 ± 19.5	0.003
Primary liver disease, n (%)					
Viral hepatitis	7 (7.6)	0	2 (6.7)	5 (16.1)	0.047
Autoimmune	24 (26.1)	5 (16.1)	10 (33.3)	9 (29.0)	0.28
MAFLD	28 (30.4)	11 (35.5)	7 (23.3)	10 (32.3)	0.57
Alcohol cirrhosis	20 (21.7)	10 (32.3)	8 (26.7)	2 (6.5)	0.03
Storage disorder	2 (2.2)	0	0	2 (6.5)	0.13
Malignancy	3 (3.3)	2 (6.5)	1 (3.3)	0	0.36
Other	8 (8.7)	3 (9.7)	2 (6.7)	3 (9.7)	0.89
Severity of liver disease					
CPT-score	8 ± 2	9 ± 2	8 ± 2	7 ± 2	0.015
MELD score	15 ± 6	16 ± 6	16 ± 4	13 ± 7	0.14
Complications of liver disease, n (%)					
Mortality on wait-list	18 (19.6)	6 (19.4)	10 (33.3)	2 (6.5)	0.03
Hepatic encephalopathy	48 (52.2)	19 (61.3)	20 (66.7)	9 (29.0)	0.006
Use of lactulose	40 (43.5)	16 (51.6)	16 (53.3)	8 (25.8)	0.051
Ascites on admission	60 (65.2)	22 (71.0)	24 (80.0)	14 (45.2)	0.012
Varices on admission	70 (76.1)	23 (74.2)	25 (83.3)	22 (71.0)	0.503
Variceal bleeding	23 (25.0)	7 (22.6)	11 (36.7)	5 (16.1)	0.167
Hepatocellular carcinoma	27 (29.3)	7 (22.6)	6 (20.0)	14 (45.2)	0.064
Hepatorenal syndrome	19 (20.7)	8 (25.8)	7 (23.3)	4 (12.9)	0.409
Medication, n (%)					
Statins	14 (15.2)	5 (16.1)	5 (16.7)	4 (12.9)	0.937
Antihypertensives	63 (68.5)	21 (67.7)	21 (70.0)	21 (67.7)	1.000
Glucose-lowering drugs	27 (29.3)	5 (16.1)	9 (30.0)	13 (41.9)	0.085
Proton pump inhibitors	58 (63.0)	22 (71.0)	18 (60.0)	18 (58.1)	0.588
Vitamin K antagonists	8 (8.7)	3 (9.7)	3 (10.0)	2 (6.5)	0.906

Data are represented as the mean ± SD, median (interquartile range) or n (%). One-way ANOVA and Fisher’s exact test were conducted for continuous and categorical variables, respectively. Abbreviations: BCAA, branched chain amino acids; BMI, body mass index; SBP, systolic blood pressure; DBP, diastolic blood pressure; CRP, C-reactive protein; AST, aspartate aminotransferase; ALT, alanine aminotransferase; ALP, alkaline phosphatase; MELD, Model for End-stage Liver Disease; CPT-score, Child Pugh Turcotte score; MAFLD, metabolic dysfunction-associated fatty liver disease.

**Table 2 biomolecules-13-00824-t002:** Plasma BCAA and alanine concentrations of the screening population with cirrhosis according to CPT categories.

Amino Acid Titles (µmol/L)	Overall (n = 92)	CPT A (n = 22)	CPT B (n = 46)	CPT C (n = 24)	*p*-Value
Total BCAA	276.0 (269.7–320.6)	364.5 (319.9–416.9)	262.5 (239.8–291.6)	248.5 (215.1–354)	<0.001
Valine	142.0 (141.2–169.1)	193.5 (172.0–225.5)	128.0 (123.7–148.8)	138.0 (112–190.4)	<0.001
Leucine	83.5 (80.4–101.3)	113.0 (94.8–128.3)	75.5 (69.4–94.1)	76.5 (59.8–118.7)	0.015
Isoleucine	48.0 (45.3–53.2)	58.5 (49.0–67.8)	44.5 (42.1–53.2)	48.0 (37.7–50.3)	0.037
Alanine	277.7 (270.5–328.3)	354.5 (298.6–398.1)	270.5 (261.3–335.6)	227.5 (220.5–319.5)	0.088

Data are represented as the median (IQ range). Differences were tested with one-way ANOVA. Abbreviations: BCAA, branched chain amino acids; CPT, Child Pugh Turcotte.

**Table 3 biomolecules-13-00824-t003:** Physical exercise tests in relation to tertiles of plasma total BCAA concentrations.

	Overall	T1	T2	T3	*p*-Value
Hand grip strength (kg)					
Subjects n = 77	32.2 ± 10.4	30.0 ± 8.6	32.2 ± 10.1	34.0 ± 11.8	0.370
Men n = 49	38.0 ± 7.9	34.0 ± 7.2	38.8 ± 5.1	40.7 ± 9.3	0.038
Women n = 28	22.1 ± 5.2	22.9 ± 5.9	19.9 ± 2.8	23.1 ± 5.9	0.377
4 m walking test (s)					
Subjects n = 42	3.9 ± 1.3	4.4 ± 1.5	3.7 ± 1.1	3.6 ± 1.1	0.242
Men n = 26	3.7 ± 1.1	3.9 ± 1.0	3.5 ± 0.8	3.7 ± 1.4	0.821
Women n = 16	4.2 ± 1.6	5.0 ± 1.9	4.0 ± 1.6	3.4 ± 0.6	0.208
Sit-to-stand test (s)					
Subjects n = 40	14.3 ± 4.7	15.5 ± 2.9	16.4 ± 6.5	12.2 ± 4.0	0.043
Men n = 27	14.2 ± 4.5	15.5 ± 3.3	15.4 ± 5.4	12.0 ± 4.3	0.159
Women n = 13	14.5 ± 5.3	15.3 ± 2.3	20.2 ± 11.7	12.4 ± 3.8	0.181
Timed up and go test (s)					
Subjects n = 40	8.5 (7.9–9.8)	8.5 (8.0–12.9)	9.0 (8.1–9.7)	7.6 (6.6–8.4)	0.043
Men n = 25	8.5 (5.6–11.4)	9.3 (7.2–11.4)	8.9 (6.7–10.2)	7.5 (5.6–10.4)	0.159
Women n = 15	9.5 (5.6–22.3)	12.5 (7.4–22.2)	8.7 (8.2–9.0)	7.6 (5.6–9.9)	0.181
Standing balance test (points)					
Subjects n = 36	2.0 ± 2.6	1.8 ± 1.9	2.4 ± 2.8	1.9 ± 3.0	0.850
Men n = 22	1.7 ± 2.4	1.8 ± 1.7	1.4 ± 1.5	1.8 ± 3.8	0.945
Women n = 14	2.5 ± 2.8	1.8 ± 2.5	4.0 ± 4.1	2.0 ± 2.0	0.487
Clinical Frailty Scale (0–9)					
Subjects n = 80	4.1 ± 1.5	4.2 ± 1.6	4.2 ± 1.7	3.8 ± 1.2	0.297
Men n = 52	4.0 ± 1.6	4.5 ± 1.9	3.5 ± 1.3	3.9 ± 1.6	0.222
Women n = 28	4.1 ± 1.2	4.2 ± 1.2	4.2 ± 1.2	4 ± 1.4	0.908

Data are represented as the mean ± SD or the median (IQ range). Differences were tested with one-way ANOVA.

**Table 4 biomolecules-13-00824-t004:** Univariate relationships of total BCAA, valine, leucine and isoleucine with physical exercise parameters.

	Total BCAA (µmol/L)	Valine (µmol/L)	Leucine (µmol/L)	Isoleucine (µmol/L)
Hand grip strength (kg)				
Men, n = 49	0.261	0.162	0.322 *	0.329 *
Women, n = 28	0.013	−0.040	0.010	−0.014
4 m walking test (s), n = 42	−0.245	−0.245	−0.208	−0.209
Sit-to-stand test (s), n = 40	−0.352 *	−0.337 *	−0.316 *	−0.170
Timed up and go test (s), n = 40	−0.472 **	−0.420 **	−0.420 **	−0.344 *
Standing balance test (points), n = 36	−0.068	−0.031	−0.134	0.010
Clinical Frailty Scale (0–9), n = 80	−0.098	−0.110	−0.090	−0.117

Relationships were analyzed by Pearson correlation coefficients; * *p* < 0.05; ** *p* ≤ 0.01.

**Table 5 biomolecules-13-00824-t005:** Baseline characteristics of 92 patients with cirrhosis divided into tertiles of alanine concentrations.

	Overall	T1	T2	T3	*p*-Value
Total (n)	92	31	30	31	
Men (n)	60	20	20	20	
Women (n)	32	11	10	11	
Laboratory measurements					
AST (U/L)	54.0 (43.3–83.0)	56.5 (55.0–90.5)	57.5 (54.4–108.0)	47.5 (39.4–101.6)	0.755
ALT (U/L)	39.5 (28.3–55.8)	39.5 (36.0–67.2)	42.0 (36.6–69.2)	39.5 (30.3–65.2)	0.770
ALP (U/L)	153.0 (156.7–197.3)	168.0 (161.6–235.8)	149.0 (140.0–201.6)	140.0 (122.5–202.2)	0.134
Bilirubin (μmol/L)	41.0 (18.3–86.0)	56.0 (48.5–159.2)	43.5 (39.7–125.4)	44.5 (27.9–152.5)	0.237
Ammonia (µmol/L)	68.8 ± 33.4	71.1 ± 35.6	68.6 ± 31.2	68.8 ± 33.4	0.884
Primary liver disease, n (%)					
Viral hepatitis	7 (7.5)	1 (3.2)	2 (6.7)	4 (12.9)	0.494
Autoimmune	24 (26.1)	9 (29.0)	9 (30.0)	6 (19.4)	0.575
MAFLD	28 (30.4)	8 (25.8)	8 (25.8)	12 (38.7)	0.468
Alcohol cirrhosis	20 (21.7)	6 (19.4)	9 (30.0)	5 (16.1)	0.450
Storage disorder	2 (2.2)	0	2 (6.7)	0	0.326
Malignancy	3 (3.3)	3 (9.7)	0	0	0.032 *
Other	8 (8.7)	3 (9.7)	1 (3.3)	4 (12.9)	0.432
Severity of liver disease					
MELD score	15 ± 6	16 ± 6	15 ± 5	13 ± 6	0.107
CPT score	8 ± 2	9 ± 2	8 ± 2	7 ± 2	0.090
Complications of liver disease, n (%)					
Mortality	18 (19.6)	6 (19.4)	9 (30.0)	3 (9.7)	0.157
Hepatic encephalopathy	48 (52.2)	16 (51.6)	13 (43.3)	19 (61.3)	0.309
Use of Lactulose	40 (43.5)	12 (38.7)	12 (40.0)	16 (51.6)	0.530
Ascites on admission	60 (65.2)	20 (64.5)	23 (76.7)	17 (54.8)	0.298
Varices on admission	70 (76.1)	21 (67.7)	25 (83.3)	24 (77.4)	0.608
Variceal bleeding	23 (25.0)	5 (16.1)	10 (33.3)	8 (25.8)	0.369
Hepatocellular carcinoma	27 (29.7)	6 (19.4)	7 (23.3)	14 (45.2)	0.064
Hepatorenal syndrome	19 (21.3)	8 (26.7)	5 (16.6)	6 (19.4)	0.661

Data are represented as the mean ± SD, median (interquartile range) or n (%). One-way ANOVA and Fisher’s exact test were conducted for continuous and categorical variables, respectively (* *p* < 0.05). BCAA, branched chain amino acids; AST, aspartate aminotransferase; ALT, alanine aminotransferase; ALP, alkaline phosphatase; MELD, Model for End-stage Liver Disease; CPT score, Child Pugh Turcotte score; MAFLD, metabolic dysfunction-associated fatty liver disease.

## Data Availability

The data presented in this study are available on request from the corresponding author.

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
