# Peer review of "Branched Chain Amino Acids Are Associated with Physical Performance in Patients with End-Stage Liver Disease"

_biomolecules, 2023, doi:10.3390/biom13050824_

Round 1

Reviewer 1 Report

The comments have been attached in the document uploaded. 

Author Response

Thank you for your review of our paper on BCAAs and physical performance. We appreciate your suggestions and comments, which will certainly help to improve the quality of our work.

We have carefully considered your feedback and have made the suggested changes to the manuscript; we believe that based on your comments these changes have improved our manuscript and have strenthened the conclusions drawn.

Reviewer 2 Report

all in all it was a pleasure to read your ms. it confirms that BCAAs in liver disease deserve renewed attention. your focus is on them as 'biomarkers' which is sound and interesting enough. it is good you briefly mention the mechanistic/pathogenic importance of low BCAAs which may be at least as intriguing.

Author Response

We thank the reviewer for taking the time to review our manuscript and for his/her positive feedback, we are pleased to hear that our  paper is interesting and informative.

We agree that BCAA in liver diseases deserve renewed attention, and we are glad that our focus on them as biomarkers sounds interesting. We also acknowledge the importance of briefly mentioning the mechanistic/pathogenic importance of low BCAAs, which may be at least as intriguing. We will ensure that this aspect is given due attention in the final version of the manuscript.

Reviewer 3 Report

Dear author:

The Fischer and BCAA to tyrosine ratios have been examined as prognostic predictors. How about their relationships with patients' characteristics and physical performances?

Also, please use multivariate analysis to compare the prediction of physical performances between plasma BCAA levels, CPT scores, and MELD scores. 

Author Response

We thank the reviewer for his/her feedback and important suggestions regarding our manuscript. We appreciate his/her interest in our research and agree that examining the relationships between the Fischer and BCAA to tyrosine ratios with patients' characteristics and physical performances would be an important area of investigation.

Regarding the suggestion to use multivariable analysis to compare the prediction of physical performances between plasma BCAA levels, CPT scores, and MELD scores, we agree that this would be a valuable addition to the paper, and we will incorporate this analysis into our final version.

Round 2

Reviewer 1 Report

My comments have been carefully responded by the authors. They have extensively improved the paper after this revision. However, I am not completely convinced by this revised manuscript. Further concerns are as follows:

1.  I agree that it's important to look into the relationship between BCAA concentrations and ESLD, which is the common illness among the enrolled patients. It was a good idea to recheck the relevance between BCAA and the three comorbidities listed in table S-1. Apparently, diabetes was an interference factor for the former analysis. It's possible that the major patients enrolled in the study suffered from MAFLD. It makes sense that these patients were in high risk of diabetes. Unfortunately, this point was not discussed in the manuscript. Besides, the variations of BCAA in the patients with MAFLD are obviously different from those with malignancy due to the Warburg effect, which is also worth discussing.   

2.  The authors state that they have validated the BCAAs by correlation coefficients between the results of NMR and LC-MS2. This step is odd and needs to be detailed (such as the manufacturers of spectrometer, the column, the mobile phase, positive/negative mode, CE, etc.).  Given that the identification of BCAAs in NMR are generally known (Iso: 0.93t, 0.99d; Leu: 0.95dd; Val: 1.03d) and the reproducibility of NMR is better than LC-MS. Neither the qualification nor the quantification needs to be validated by LC-MS/MS.

Author Response

We thank the reviewer for taking the time to review our revised manuscript and for the feedback. We appreciate that he/she acknowledge the authors' efforts in addressing the initial comments and making extensive revisions to the paper; the authors have reviewed the additional comments and have responded to them as shown below.

In the revised manuscript, we have added a sentence in the Introduction section (final paragraph), now more clearly stating the central hypothesis of our study:  We hypothesized that low circulating BCAA concentrations could be associated with poor physical performance in ESLD.
